# Hadiths Classification Using a Novel Author-Based Hadith Classification Dataset (ABCD)

Ahmed Ramzy [1], Marwan Torki [2], Mohamed Abdeen [3], Omar Saif [4,5], Mustafa ElNainay [6], AbdAllah Alshanqiti [3] and Emad Nabil [1,3,*]

[1] Faculty of Computers and Artificial Intelligence, Cairo University, Giza 12613, Egypt; a.ramzy@fci-cu.edu.eg
[2] Department of Computer and Systems Engineering, Alexandria University, Alexandria 21526, Egypt; mtorki@alexu.edu.eg
[3] Faculty of Computer and Information Systems, Islamic University of Madinah, Madinah 42351, Saudi Arabia; mabdeen@iu.edu.sa (M.A.); amma@iu.edu.sa or a.m.alshanqiti@gmail.com (A.A.)
[4] Faculty of Hadith and Islamic Studies, Islamic University of Madinah, Madinah 42351, Saudi Arabia; osaif@iu.edu.sa or osaif@alqasimia.ac.ae
[5] Faculty of Sharia and Islamic Studies, Alqasimia University, Sharjah 63000, United Arab Emirates
[6] Faculty of Computer Science and Engineering, Alamein International University, New Alamein City 51718, Egypt; melnainay@aiu.edu.eg
* Correspondence: emadnabil@iu.edu.sa or emadnabilcs@gmail.com or e.nabil@fci-cu.edu.eg

**Abstract:** Religious studies are a rich land for Natural Language Processing (NLP). The reason is that all religions have their instructions as written texts. In this paper, we apply NLP to Islamic Hadiths, which are the written traditions, sayings, actions, approvals, and discussions of the Prophet Muhammad, his companions, or his followers. A Hadith is composed of two parts: the chain of narrators (Sanad) and the content of the Hadith (Matn). A Hadith is transmitted from its author to a Hadith book author using a chain of narrators. The problem we solve focuses on the classification of Hadiths based on their origin of narration. This is important for several reasons. First, it helps determine the authenticity and reliability of the Hadiths. Second, it helps trace the chain of narration and identify the narrators involved in transmitting Hadiths. Finally, it helps understand the historical and cultural contexts in which Hadiths were transmitted, and the different levels of authority attributed to the narrators. To the best of our knowledge, and based on our literature review, this problem is not solved before using machine/deep learning approaches. To solve this classification problem, we created a novel Author-Based Hadith Classification Dataset (ABCD) collected from classical Hadiths' books. The ABCD size is 29 K Hadiths and it contains unique 18 K narrators, with all their information. We applied machine learning (ML), and deep learning (DL) approaches. ML was applied on Sanad and Matn separately; then, we did the same with DL. The results revealed that ML performs better than DL using the Matn input data, with a 77% F1-score. DL performed better than ML using the Sanad input data, with a 92% F1-score. We used precision and recall alongside the F1-score; details of the results are explained at the end of the paper. We claim that the ABCD and the reported results will motivate the community to work in this new area. Our dataset and results will represent a baseline for further research on the same problem.

**Keywords:** machine learning; deep learning; classification; Hadith classification; religious studies; Natural Language Processing

## 1. Introduction

A Hadith is one of the most famous sources of classical Arabic texts. It comprises narrative sayings, actions, approvals, and discussions of the Prophet Muhammad, his companions, or followers. For Muslims, a Hadith is the second most important source of religious instructions after the Qur'an.

Each Hadith is composed of two components: the actual wording of a Hadith text, known as Matn, and the chronological list of people who transmitted the Matn, which is





known as Sanad. The Sanad (or the narration chain) consists of narrators, each narrator mentioning the one from whom he heard the Hadith.

Over the last few decades, researchers in the field of NLP, especially in the Arabic language, have become more interested in Hadiths and related research. We can group research efforts primarily into two groups: content-based research and narration-based research. The content-based research focuses on Hadith Matn. The narration-based research focuses on Hadith Sanad and its narration chain. The works of [1,2] have presented various research studies on Hadiths in detail.

Works that processed the actual Hadith text (Matn) can be categorized into information retrieval, ontology, query expansion, and question-answering system, topic segmentation, and classification of Hadiths.

Studies that processed the narration chain of Hadiths (Sanad) can be grouped into Hadith authentication or veracity, visualization narration graph, and disambiguation of a narrator's name. Disambiguation problems arose as there were lots of different forms/spellings for the same narrator [3].

Hadith classification, which is the subject of this paper, can be categorized from different perspectives, as explained in Figure 1. Hadith can be classified into a specific chapter or topic of Sahih Al-Bukhari's book [4], such as knowledge, faith, hajj, praying, etc. Sahih Al-Bukhari's book is considered one of the most trusted compilations of Hadiths.

A Hadith can also be classified according to its authentication or veracity, such as Sahih, Hassan, Daif, and Mawdoa. Hadith authentication mainly depends on Sanad; however, some studies used the Matn in the classification.

Finally, Hadiths can be classified as Marfu, Mawquf, and Maqtu, according to the origin of a narration. This last classification perspective specifies whether a Hadith is attributed to Prophet Muhammad, a companion (Sahabi), or a successor (Tabi'i). Marfu class refers to a narration attributed specifically to the Prophet Muhammad. Mawquf refers to a narration attributed to the companion (Sahabi) and Maqtu refers to a narration attributed to the successor of one of Muhammad's companions (Tabi'i).

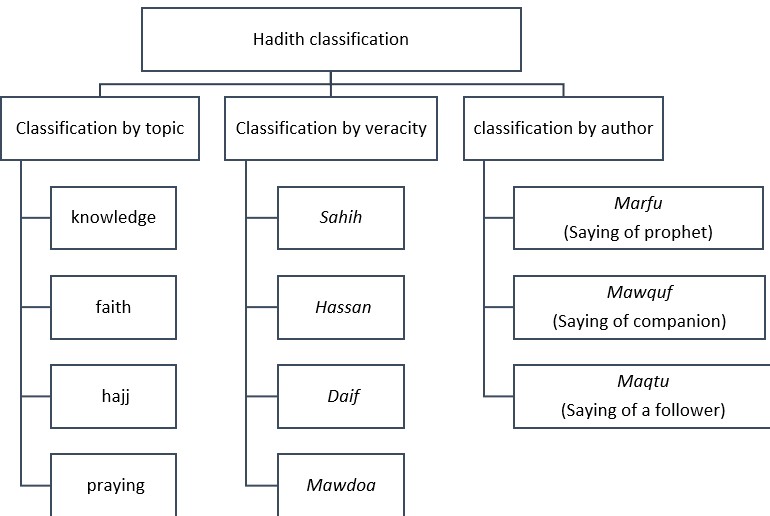

**Figure 1.** Hadith classification types.

Hadith classification is still an open problem in Islamic Hadith studies for the following reasons. There are written transcripts that were discovered over time, and Islamic scholars need to know the author of this text. Also, some current Hadiths are not clear about who their author is. Therefore, scholars are not sure who the authors are. Moreover, it is not indeed that the author is at the end of the narration list. For these reasons, we chose to produce an artificial intelligence-based (AI-based) solution for this problem.

In this study, we used a novel approach to classify Hadiths according to the author or the origin of narration. We used three training data types, namely Sanad alone, Matn alone,

and both Sanad and Matn. We explored machine learning and deep learning techniques. Combining Matn and Sanad produced poor results, as they have totally different meanings, so we excluded these results from the Results section.

As with every machine/deep learning algorithm, it is hard to achieve 100% accuracy, so here, we propose a tool that can be used as decision support for Hadith scholars to guide them or provide a Hadith classification opinion.

For the sake of training, we have created a novel structured Hadith dataset. The dataset passed lots of preprocessing to be ready for usage, like the separation of Matn and Sanad. In addition, we parsed the narrators in the Sanad and performed normalization for the identification of the narrator's name to avoid ambiguity in identifying the narrators in Sanad. The dataset is released to the community for scientific use. A detailed description of the dataset is explained in a separate section of the paper.

The main contribution of this paper is a proposed benchmark dataset that can be used by the research community to classify a Hadith according to its origins, such as Marfu, Mawquf, and Maqtu, in addition to introducing a baseline model for this classification problem using machine learning and deep learning techniques.

The rest of this paper is organized as follows. The literature review is discussed in Section 2. The description of the dataset is presented in Section 3. Detailed key contributions of the paper, proposed methodologies, and experimental results are presented in Section 4. Section 5 discusses the experimental evaluation. Finally, Section 6 concludes the research with a discussion and an outlook on future work. For the sake of convenience, all terminologies and special words mentioned in this paper are briefly explained at the end of the paper, in the Glossary.

## 2. Literature Review

Hadith classification is the process of classifying Hadiths into predefined classes. We grouped all the research that is related to Hadith classification. A Hadith can be classified from different perspectives. Hadiths can be classified by topic, authenticity, or by author, as illustrated in Figure 1. In this section, we explain the related work in each.

Based on our extensive literature review, it appears that no machine/deep learning approaches have been employed to solve the classification problem at hand. To address this gap, we developed a novel dataset called the Author-Based Hadith Classification Dataset (ABCD), which we gathered from classical Hadith books. The ABCD contains a comprehensive collection of 29,000 Hadiths, and it is unique in that it includes information of 18,000 distinct narrators. This wealth of information will enable researchers to delve deeper into the characteristics of these narrators and their relationship to the Hadiths they have transmitted.

The ABCD dataset is a valuable resource for scholars and researchers in the field of Hadith studies. By utilizing machine/deep learning approaches, we aim to provide a more accurate and efficient classification of Hadiths based on their authors. Additionally, the ABCD dataset could serve as a benchmark for future studies in the field of Hadith classification.

### 2.1. Classification by Topic

The authors in [5–7] classified Hadiths according to their content (Matn) and assigned each Hadith a topic from a set of predefined topics. The topics included faith, knowledge, judgment, and crimes. These topics are extracted from the famous Hadith book, Sahih Al-Bukhari.

The authors in [8] classified Hadiths using Matn, but from a different angle: the classification classes were the meaning of the Matn, e.g., order for doing something, forbidding doing something, or just information without any order. The authors in [9] have proposed a semantic similarity metric that could also be applied to the Hadith classification problem.

## 2.2. Classification by Authenticity

Other researchers classify Hadiths according to their authenticities, such as Sahih, Hassan, Daif, and Mawdoa. Hadith authenticity has some equivalent terms like Hadith veracity, Hadith judgment, or grading the Hadith; all of them mean the same thing.

Typically, Muslim scholars classify a Hadith to determine if it is authentic or not. There are frequent criteria for determining authenticity. These criteria include the Sanad's chain of narrators and the number of narrators involved in each level in the Sanad chain [10]. You need to be well-versed in Hadith science to evaluate a Hadith's authenticity.

This is a susceptible and significant task; it should only be performed by a Hadith expert. However, there are some studies that were conducted to build an expert system for this type of classification to help non-experts understand what the rules behind the classification are.

Hadith authenticity mainly depends on the Sanad; however, there are some works that depend on the Sanad in addition to the Matn [11,12]. The reported authenticity classification experiments have been evaluated using different datasets.

The authors in [13] used association rules to classify Hadiths as Sahih, Hassan, Daif, and Mawdoa, depending on the accuracy of the narrators and the continuity of Sanad. In [14], the authors used the decision tree and Naïve Bayes to classify the Hadith authenticity. They tested the classifier on 999 Hadiths. The decision tree slightly outperformed the Naïve Bayes classifier.

In [15,16], the authors implemented domain-dependent ontology to help in authenticity assessment. The objective of building the ontology is to improve the features of the Sanad with more links, properties, and relations. Their model classifies the Hadith as Sahih, Hassan, and Daif. They picked 16 Hadiths from Sunan Ibn Majah's book [16] to test their model.

In [17], the authors proposed a heuristic-based approach with no machine learning algorithms to determine the Hadith authenticity. They evaluated their approach on 3000 Hadiths from Sahih Al-Bukhari and Sunan at-Tirmizi books [18].

## 2.3. Classification by Author

As we mentioned before, the classification of Hadiths, according to the author, can be classified as Hadith Marfu, Hadith Mawquf, and Hadith Maqtu. This classification is particularly important since it allows readers to distinguish the Prophet's sayings from his companions or successors at a glance; this is especially beneficial in Fiqh debates.

To our knowledge, there is no research work that has reported a model which classifies this type of classification nor introduced a dataset that contains labeled Hadiths, nor narrators' datasets with their attributes. The created dataset is not only used in classification by the author, but can also be used in Hadith classification by authenticity as the dataset contains detailed data about narrators, which is important in authenticity classification.

The only found research that worked in the classification by the author is reported in [19], where the authors used a rule-based approach to classify Hadiths. They employed service-oriented architecture to overcome the communicational problem and a candidature for the Software as a Service (SaaS) for cloud computing. They did not report the dataset, accuracy, or performance measures of their model.

## 3. The Author-Based Hadith Classification Dataset (ABCD)

After surveying the literature, we found that there is no dataset suitable for the Hadith classification problem, so we created a new dataset from scratch.

We selected Hadiths from different Hadith collection books to ensure that the dataset represents a diverse and balanced set of Hadith classes. The selection of Hadiths was based on their relevance to the Hadith classification problem and their inclusion in well-known Hadith collections.

In addition, we included information about the narrators in the dataset, such as their attributes, personal information, and different name forms, as this information can be useful for researchers interested in investigating the role of narrators in Hadith classification.

Regarding the size of the dataset, we aimed to create a balanced dataset that is representative of the broader body of Hadith literature. We believe that the size of the dataset (29 K Hadiths and 18 K narrators) is appropriate for this purpose. Indeed, the bigger the dataset, the better results we can get, so, in future work, we intend to release a bigger extended dataset, but for now, the ABCD can be considered as a good starting point for Hadith classification. Furthermore, we plan to release the dataset for free to the community to facilitate further research in this area.

Overall, we believe that our dataset selection criteria and creation process, which involved identifying Hadith classes and selecting suitable Hadith books to ensure balance, provide a solid foundation for research on Hadith classification.

Figure 2 illustrates the methodology we followed in the dataset creation and the phase of the experiment. Our model classifies Hadiths based on the Hadith's author. Hence, our class labels are classified as Marfu, Mawquf, or Maqtu. We chose three suitable Hadith collection books [19–21] to make the collected Hadiths balanced in terms of classes. We collected around 29 K. Each Hadith contains two parts, which are its Sanad and its Matn. Every Hadith is divided into Sanad and Matn. The preprocessing challenges for each part have been handled separately. For the Matn, we relied on the CAMeL tool [22] to perform cleanings, such as removing the punctuations, diacritics, special characters, non-Arabic characters, non-Unicode characters, etc. For the Sanad part, first, we separated the narrators of each Sanad and saved them in a chronological order as they appeared in the Sanad.

Every narrator may have different name forms mentioned in Hadiths. We collected the whole name forms for each narrator, then performed a normalization process to assign one identifier to every narrator, considering their different name forms. At this moment, each Hadith has its class label, preprocessed Matn, and normalized Sanad with a chronological order of a unique identifier for each narrator.

Finally, we created a balanced dataset of Hadiths that the research community can use for scientific purposes. The creation process of the dataset is explained as follows:

- Identifying Hadith classes;
- Selecting suitable Hadith books for collection Hadiths, so the Hadith classes are balanced;
- Every collected Hadith is divided into the Sanad and Matn;
- Every Sanad is normalized;
- Every Matn is cleaned with the help of the CaMEL tool;
- Storing the processed Sanad and Matn in a database.

All phases illustrated in Figure 2 are explained thoroughly throughout the paper.

Figure 3 explains the terms Matn, Sanad, and the Hadith narrators by example from Mosnad Imam Ahmad's book. The first single underlined part is called Sanad, and as illustrated, it contains five highlighted narrators. The second part (double underlined) contains the Matn, which is the actual content of the Hadith. The book where this Hadith is mentioned, together with its number, is described in line number three.

To facilitate the usage of the ABCD, we split the Matn from the Sanad in addition to separating the narrators of the Sanad. One of the challenges that we faced is that the Sanad may contain a name, and this name refers to different persons in different Sanads. In other words, this name may appear in two Sanads, and it refers to two different persons, as explained in Figure 4. There is an example of ambiguity in narrators. The name Hammad in Sanad 1 and Sanad 2 refers to two different persons [23]. Thus, normalization was mandatory to remove the ambiguity of narrators.

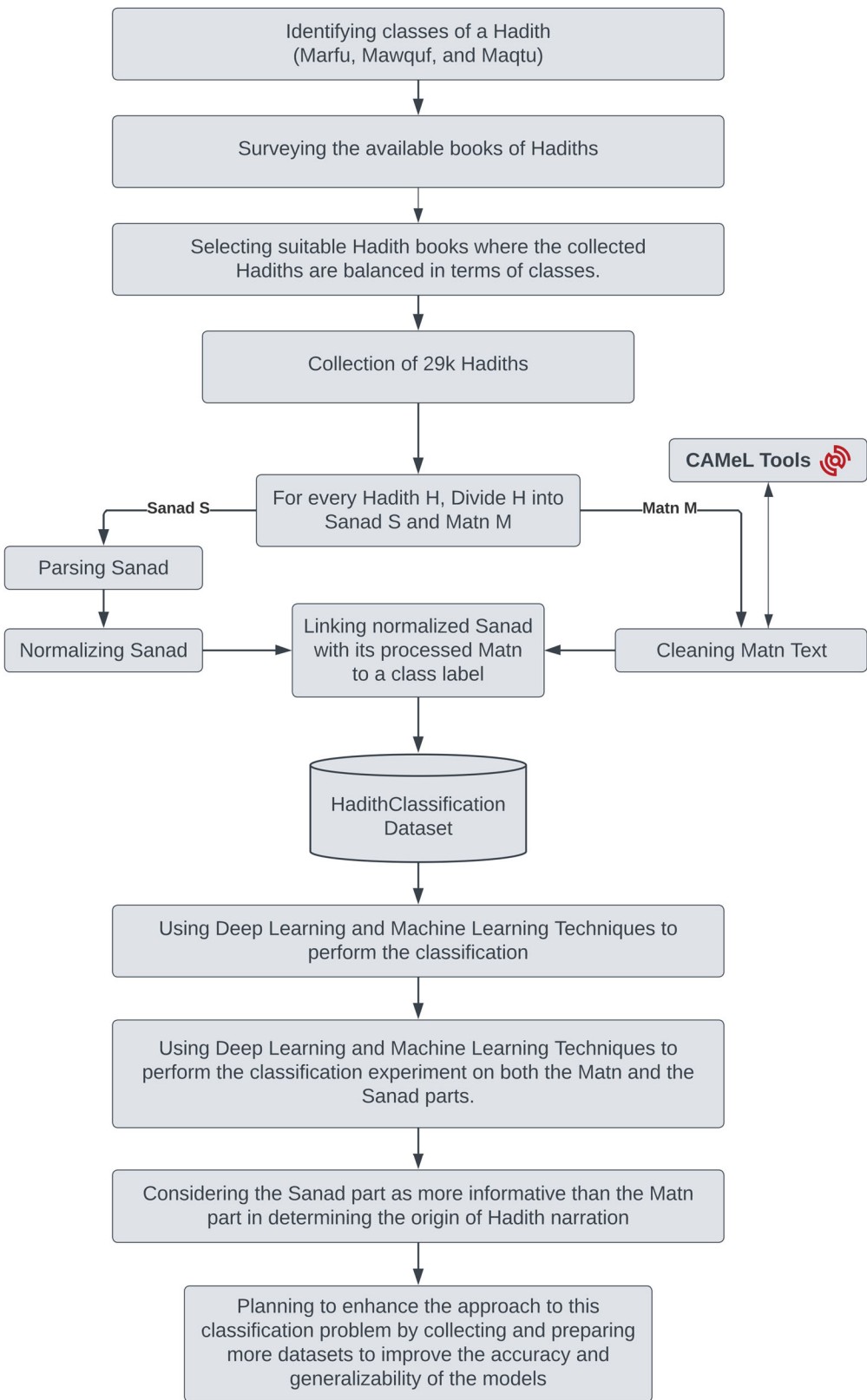

**Figure 2.** Methodology of the dataset creation and experiments.

حَدَّثَنَا مُوسَى بْنُ دَاوُدَ ، حَدَّثَنَا عَبْدُ اللهِ بْنُ عُمَرَ ، عَنِ ابْنِ شِهَابٍ ، عَنْ عَلِيٍّ بْنِ حُسَيْنٍ ، عَنْ أَبِيهِ قَالَ : قَالَ رَسُولُ اللهِ صَلَّى اللهُ عَلَيْهِ وَسَلَّمَ :

مِنْ حُسْنِ إِسْلَامِ الْمَرْءِ تَرْكُهُ مَا لَا يَعْنِيهِ .

مسند أحمد بن حنبل - رقم 1761

**Figure 3.** An example of a Hadith structure with the Sanad and Matn written in Arabic. The Matn is highlighted where the Sanad is double underlined.

**Figure 4.** An example of the ambiguity of narrators.

We mainly relied on [24] to collect Hadiths. We collected 29 K Hadiths from three different Hadith collection books. We obtained around 21 K Hadiths from Musannaf of Abd Al-Razzaq's book [20], around 4 K Hadiths from Sunan ibn Majah [19], and about 4 K Hadiths from Sunan al-Darimi [21]. There are three labels or classes for the collected Hadiths, which are Hadith Marfu, Hadith Mawquf, and Hadith Maqtu. The reason behind choosing the previous three books among many other books is to make our dataset balanced in terms of classes or labels.

The dataset contains 29 K Matn and 18 K unique Sanads. The reason that the two numbers are not the same is that the same chain of narrators may narrate more than one Matn. For every narrator in the 18 K Sanads, we collected some essential attributes. Some of these attributes helped for the narrator normalization. We did not use all of these attributes, but we included them in the dataset for future work and other possible usages of the dataset by the community.

The narrator's attributes are as follows:

1. Narrator's nickname and all the different name forms of his name;
2. Narrator's descent;
3. Narrator's relationships;
4. Narrator's narrations;
5. Narrator's date of birth;
6. Narrator's country of birth;
7. Narrator's country of residence;
8. Narrator's visited cities and when he visited them;
9. Narrator's date of death;
10. Narrator's country of death;
11. Narrator's school of thought, also called Madhhab [25];
12. Narrator's rank according to Ibn Hajar [26];
13. Narrator's rank according to Al-Zahabi [27];
14. Narrator's rank according to Taqreeb Al-tahzeeb [28].

Another challenge we faced is that a single narrator's name may have more than one spelled form in different Sanads. For example, a narrator named "Abdullah Ibn Abbas" may be mentioned in another Hadith with "Ibn Abbas". Also, any narrator whose name starts with "Abu" may also be mentioned starting with "Aba" or "Abi" according to Arabic grammatical rules. In the same context, the narrator's name "Amro: عمرو" may appear like "Amra: عمرا" according to the grammatical rules too. Table 1 shows an example of

this problem. As such, the dataset underwent a normalization process so that the different appearing forms of a narrator are indexed, and all the appearing forms of each narrator are represented using only one identification number (ID). We have returned to [24] in order to collect the narrator's names and their different spelling forms. Figure 5 shows a complete example of converting Sanad to a chain of identifiers. Therefore, we performed preprocessing on the Sanad part of the Hadiths to ensure an accurate identification of the narrators. Specifically, the narrators in the Sanad were parsed and all of the different name forms for each narrator collected. Then, a normalization process was performed to assign a unique identifier to each narrator, taking into account their different name forms.

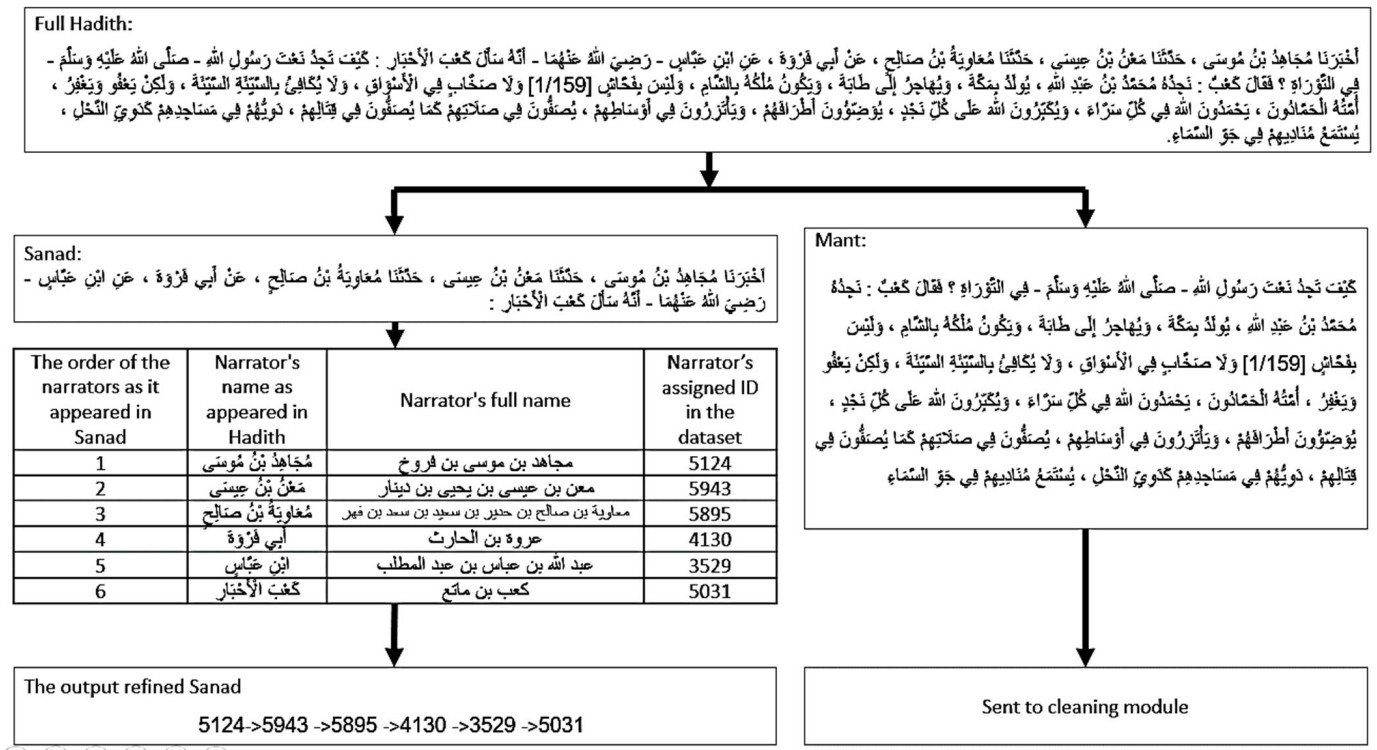

**Figure 5.** An illustration of a complete example of normalizing narrators from name forms to unique identifiers.

This normalization process likely involved mapping each unique name form to a single identifier, which can be used to distinguish between different narrators with similar or identical names. By assigning a unique identifier to each narrator, it was ensured that the classification models accurately capture the relationships between the narrators in the Sanad. Additionally, the chronological order of the unique identifier can help with understanding the chain of narration and identifying the authenticity of the Hadiths.

Overall, the normalization process performed on the Sanad part of the Hadiths helps address the challenge of ambiguity in identifying the narrators, which can be particularly challenging given that each narrator may have different name forms mentioned in the Hadiths. By normalizing the Sanad and assigning a unique identifier to each narrator, the relationships and authenticity of the Hadiths were capture more accurately. We mainly relied on [24] for the narrator attributes, to ensure that each unique name form in each Hadith was mapped to a unique identifier for each narrator based on narrator attribute values other than the appearing or the spelling forms, which may change from one Hadith to another.

The Matn part also required some preprocessing. We used the CAMel tool [22] to perform the cleaning preprocessing on the Matn part in order to optimize the Matn part of the Hadith for classification. We performed a series of preprocessing steps to clean the text.

Specifically, we removed various types of characters and symbols that may be irrelevant or disruptive to the classification process. These included:

1.  Punctuation: We removed all punctuation marks, such as commas, periods, and question marks, as they do not contribute to the meaning of the text and may interfere with the classification process.
2.  Diacritics: We removed all diacritical marks from the text, which are small symbols placed above or below letters to indicate pronunciation. While these marks are important for reading and understanding Arabic text, they can be ignored for the purposes of text classification.
3.  Special characters: We removed any special characters or symbols that are not part of the standard Arabic alphabet.
4.  Non-Arabic characters: We removed any characters that are not part of the Arabic script, as they are not relevant to the classification process.
5.  Non-Unicode characters: We removed any characters that are not part of the Unicode standard, which is used to represent text in a standardized format across different languages and platforms.

By removing these characters and symbols, we were able to optimize the Matn part of the Hadiths for classification. This preprocessing step ensured that the text was consistent and standardized, which can improve the accuracy and reliability of the classification models.

Overall, the preprocessing steps performed on the Matn part of the Hadiths involved cleaning the text by removing various types of characters and symbols that may be irrelevant or disruptive to the classification process. This ensures that the Matn is optimized for classification and can be accurately classified based on its content. Figure 6 illustrates an example of Hadith Matn before and after cleaning.

**Table 1.** Appearance count of different name forms for a given narrator in Arabic language.

| Narrator's Name Form | Appearance Count |
| --- | --- |
| ابن عباس | 21,956 |
| عبد الله بن عباس | 771 |
| عبد الله | 22 |
| عبد الله بن العباس | 18 |
| ابن العباس | 6 |
| أبو عباس | 3 |
| أبو الفضل يعني عبد الله بن عباس | 1 |
| ابن عم نبيكم ابن عباس | 1 |
| العباس | 1 |
| عباس | 1 |

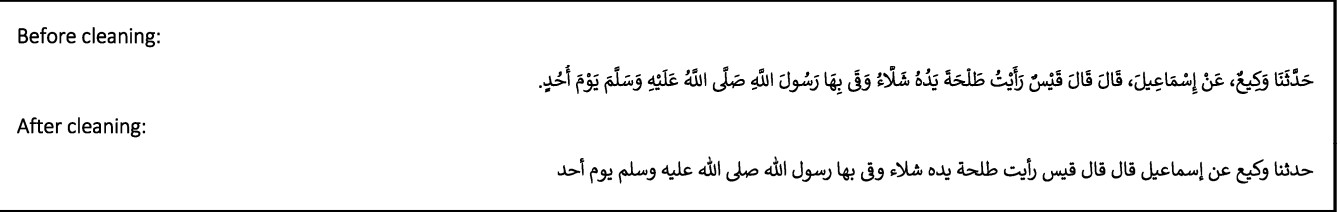

**Figure 6.** Matn of a Hadith before and after cleaning in Arabic language.

Regarding the ethics of the ABCD creation, we completely understand the importance of ethical considerations when dealing with religious texts, and we have taken several steps to ensure the respectful and sensitive handling of these texts in our research.

First, we consulted with domain experts who have a deep understanding of Hadith literature and Islamic culture to ensure that our research aligns with their views and does not misinterpret or misclassify the Hadiths. We also asked for feedback from the same experts to validate our results and ensure that they align with the religious teachings and principles.

We also made sure that our research focused solely on the classification of Hadiths and did not involve any interpretation or analysis of the religious teachings or principles.

Furthermore, we made sure to use appropriate language and terminology throughout our research to avoid any potential misinterpretations or misunderstandings.

Moreover, as we mentioned before, we relied on a trusted source such as [24] to collect the Hadiths and the narrators' attributes.

Overall, we have taken several measures to ensure the respectful and sensitive handling of religious texts in our research.

## 4. Experimental Results

We used deep learning in addition to classical machine learning approaches to tackle the Hadith classification problem. Our methodology is investigating classification performance using Matn, then using Sanad. Hence, the classification problem was solved four times. Classification was performed via ML using Matn and Sanad, followed by classification via DL also using Matn and Sanad. The best classification result will be our baseline. Then, the community can use the dataset and baseline for further enhancement.

Regarding the deep learning approach, we investigated the effectiveness of two powerful DL methods, namely RNN and Transformer-based classifiers. On the other hand, we used Gaussian NB, logistic regression, SVM, and MLP from the ML repertoire. The reason behind choosing the latter methods is that they proved their superiority in many applications, and at the same time, they have a different way of classification, except MLP and logistic regression. In a balancing mode, we divided the target labels of our dataset into 70% training and 30% validating, as usual, which gave us 21,970 training examples in total and 5491 examples for validation. We used Keras [29] and Scikit Learn [30] for implementing our models, alongside the CAMeL tool [22] for text cleaning. All experiments were run on an intel-i9 machine that supports NVIDIA 3070 GPU.

### 4.1. Performance Metrics

In the experiments, we employed standard performance metrics to contrast the various approaches. Specifically, precision (as expressed in Equation (1)), recall (as expressed in Equation (2)), and the F1-score (as expressed in Equation (3)) were all utilized.

$$\text{Precision} = \frac{\text{True Positive}}{\text{True Positive } + \text{ False Positive}} \tag{1}$$

$$\text{Recall} = \frac{\text{True Positive}}{\text{True Positive } + \text{ False Negative}} \tag{2}$$

$$\text{F1-score} = \frac{2 \times \text{True Positive}}{2 * \text{True Positive } + \text{ False Positive } + \text{ False Negative}} \tag{3}$$

We chose to use the precision, recall, and F1-score because they are widely used metrics in classification tasks. To avoid any biases or limitations for these metrics, we planned for the ABCD to be a balanced dataset and a good representative of the broader body of Hadith literature.

The ABCD includes a large number of samples from different categories, which helps mitigate the potential imbalances between classes that can affect precision, recall, and the F1-score. Additionally, we carefully curated our dataset and ensured the accuracy of ground truth labels to minimize potential ambiguity in the labeling process.

By using the previously mentioned metrics, we were able to gain a more comprehensive understanding of classifiers' performance and reduce the potential for overfitting.

### 4.2. Deep Learning-Based Classification

In our research project, we experimented with different model architectures in deep learning, including different encoding and output representations, to develop effective classifiers for Hadiths. We tested various models based on recurrent neural networks (RNN) and Transformers-based classifiers, which are state-of-the-art techniques in deep learning.

The best-performing architectures for RNN and Transformers-based classifiers are presented in Tables 2 and 3, respectively. These tables provide information about the architecture of each model, including the number of layers, input and output dimensions, activation functions, and other relevant parameters.

We used common parameters between the two methods, which are presented in Table 4. These parameters include the learning rate, optimizer, loss function, and batch size.

**Table 2.** RNN–LSTM architecture parameters.

| RNN–LSTM for Sanad Features | | | RNN–LSTM for Matn Features | | |
|---|---|---|---|---|---|
| Size of vocabulary | | | | | |
| 5266 | | | 56,205 | | |
| Max sequence (to reduce the number of trainable parameters) | | | | | |
| 30 | | | 832 | | |
| Total trainable params | | | | | |
| 726,275 | | | 7,245,955 | | |
| Top main layers in the architecture | | | | | |
| Layer (type) | Output Shape | Num. of Parameters | Layer (type) | Output Shape | Num. of Parameters |
| InputLayer (TextVectoriz) | (None, 30) | 0 | InputLayer (TextVectoriz) | (None, 832) | 0 |
| Embedding Layer | (None, 30, 128) | 674,688 | Embedding Layer | (None, 832, 128) | 674,688 |
| LSTM Layers | (None, 64) | 49,408 | LSTM Layers | (None, 64) | 49,408 |
| A standard fully connected Dense Layer with kernel_regularizer (L1 and L2) | (None, 32) | 2080 | A standard fully connected Dense Layer with kernel_regularizer (L1 and L2) | (None, 32) | 2080 |
| denseOuput | (None, 3) | 99 | denseOuput | (None, 3) | 99 |

**Table 3.** Transformer architecture parameters.

| Transformer for Sanad Features | | | Transformer for Matn Features | | |
|---|---|---|---|---|---|
| Size of vocabulary | | | | | |
| 5266 | | | 56,205 | | |
| Max sequence (to reduce the number of trainable parameters) | | | | | |
| 30 | | | 832 | | |
| Number of multi-head attentions | | | | | |
| 2 | | | 2 | | |
| Total trainable params | | | | | |
| 819,747 | | | 7,339,427 | | |
| Top main layers in the architecture | | | | | |
| **Layer (type)** | **Output Shape** | **Num. of Parameters** | **Layer (type)** | **Output Shape** | **Num. of Parameters** |
| InputLayer (TextVectoriz) | (None, 30) | 0 | InputLayer (TextVectoriz) | (None, 832) | 0 |
| Embedding Layers | (None, 30, 128) | 674,688 | Embedding Layers | (None, 832, 128) | 7,194,368 |
| TransformerBlock with the following specific layers (PositionEmbedding layer + 2 layers normalization + 2 dropout layers with rate = 0.1) | (None, 30, 128) | 140,832 | TransformerBlock with the following specific layers (PositionEmbedding layer + 2 layers normalization + 2 dropout layers with rate = 0.1) | (None, 832, 128) | 140,832 |
| global_average_pooling_1D | (None, 128) | 0 | global_average_pooling_1D | (None, 128) | 0 |
| A standard fully-connected Dense Layer with kernel_regularizer (L1 and L2) | (None, 32) | 4128 | A standard fully-connected Dense Layer with kernel_regularizer (L1 and L2) | (None, 32) | 4128 |
| denseOuput | (None, 3) | 99 | denseOuput | (None, 3) | 99 |

**Table 4.** Common hyperparameters between the models mentioned in Tables 3 and 4.

| Common Hyperparameters between the above Models | |
|---|---|
| Optimizer | Adam, β1 = 0.9, β2 = 0.9, and epsilon = $1 \times 10^{-7}$ |
| loss | Categorical Crossentropy |
| Training batch_size | 100 |
| Epochs | Various runs ranged from 10 to 100, where each epoch runs in approximately 10 min. |

To evaluate the performance of the models, we trained each model for 100 epochs, with the model being saved and its performance attributes (precision, recall, and F1-score) recorded every ten epochs. Surprisingly, we found that the models did not require a large number of epochs to achieve optimal performance. In fact, the best results were achieved when the number of epochs was less than 30.

Our experimental results showed that the Sanad part of the Hadiths was more robust in classification than the Matn part. This is because the Sanad part contains information about the chain of narration, which is critical for identifying the author of the Hadith. In contrast, the Matn part contains the text of the Hadith, which may be more challenging to classify accurately.

Furthermore, we found that RNN–LSTM-based models outperformed Transformers-based models in classification accuracy. However, we also found that the performance of RNN–LSTM-based models did not improve when applied to the Matn part of the Hadiths and produced poor results. This is likely because the Matn part requires more complex models such as Transformers, which are better able to capture the intricate patterns and details in the text.

Overall, our experiments with different model architectures in deep learning demonstrated that the Sanad part of the Hadiths is more reliable in classification and that RNN–LSTM-based models are superior to Transformers-based models for classification accuracy. These findings can help guide the development of more effective classification models for Hadiths.

Table 5 provides the classification performance of the deep learning models for both Matn and Sanad parts of Hadiths. The table includes the precision, recall, and F1–score for each model, which are commonly used metrics to evaluate the performance of classification models.

**Table 5.** Classification performance measures using DL models.

| Epochs Num. | Matn | | | | | | Sanad | | | | | |
| | RNN–LSTM | | | Transformers | | | RNN–LSTM | | | Transformers | | |
| | Precision | Recall | F1-Score | Precision | Recall | F1-Score | Precision | Recall | F1-Score | Precision | Recall | F1-Score |
| --- | --- | --- | --- | --- | --- | --- | --- | --- | --- | --- | --- | --- |
| 10 | **0.13666** | **0.33333** | **0.19333** | **0.64777** | **0.67832** | **0.6627** | 0.90211 | 0.93338 | 0.91748 | 0.9012 | 0.92997 | **0.91536** |
| 20 | 0.13666 | 0.33333 | 0.19333 | 0.64158 | 0.65255 | 0.64702 | **0.90747** | **0.93636** | **0.92169** | 0.88492 | 0.91896 | 0.90162 |
| 30 | 0.13666 | 0.33333 | 0.19333 | 0.64379 | 0.65598 | 0.64983 | 0.89491 | 0.93199 | 0.91308 | **0.88825** | **0.91882** | 0.90328 |
| 40 | 0.13666 | 0.33333 | 0.19333 | 0.63959 | 0.64652 | 0.64303 | 0.898 | 0.9264 | 0.91198 | 0.88199 | 0.91403 | 0.89773 |
| 50 | 0.13666 | 0.33333 | 0.19333 | 0.62811 | 0.63965 | 0.63382 | 0.87674 | 0.91438 | 0.89516 | 0.87794 | 0.91005 | 0.8937 |
| 60 | 0.13666 | 0.33333 | 0.19333 | 0.64683 | 0.65312 | 0.64996 | 0.88551 | 0.92325 | 0.90398 | 0.87157 | 0.90814 | 0.88948 |
| 70 | 0.13666 | 0.33333 | 0.19333 | 0.64353 | 0.64722 | 0.64537 | 0.88425 | 0.92422 | 0.90379 | 0.86364 | 0.90638 | 0.88449 |
| 80 | 0.13666 | 0.33333 | 0.19333 | 0.63609 | 0.64241 | 0.63923 | 0.88037 | 0.92421 | 0.90176 | 0.87094 | 0.90347 | 0.8869 |
| 90 | 0.13666 | 0.33333 | 0.19333 | 0.636 | 0.64724 | 0.64157 | 0.87532 | 0.92054 | 0.89736 | 0.86544 | 0.89644 | 0.88067 |
| 100 | 0.13666 | 0.33333 | 0.19333 | 0.62969 | 0.63544 | 0.63255 | 0.88706 | 0.92472 | 0.9055 | 0.88337 | 0.90741 | 0.89523 |

Precision represents the proportion of true positives (correctly classified Hadiths) to the total number of predicted positives (all Hadiths classified as belonging to a particular class). The recall represents the proportion of true positives to the total number of actual positives (all Hadiths that actually belong to a particular class). The F1-score is a weighted average of precision and recall, and is often used as a combined measure of a model's accuracy.

In Table 5, the classification performance is reported separately for the Matn and Sanad parts of the Hadiths. The table shows that the best-performing models achieved high precision, recall, and F1-scores, indicating that they accurately classified the Hadiths based on their content.

Moreover, the results in Table 5 demonstrate that the deep learning models performed better on the Sanad part of the Hadiths than on the Matn part. This is consistent with the findings mentioned earlier that the Sanad part is more reliable in classification than the Matn part.

Overall, Table 5 provides a summary of the classification performance of the deep learning models used in the study, which can be useful for evaluating and comparing the effectiveness of different models in classifying Hadiths.

### 4.3. Machine Learning-Based Classification

In our research project, we used a machine learning approach to classify Hadiths based on their Matn and Sanad parts. To extract the model features, we began with the Matn part and performed text cleaning to remove any irrelevant information or noise from the text. We then represented the Matn text using the term frequency-inverse document frequency (TF-IDF) technique, which assigns weights to the words in the text based on their frequencies and relevance to the document. Additionally, we used unigrams, bigrams, and trigrams instead of using only unigrams, which added more semantics to the classification models.

Next, we used the TF-IDF representation of the Matn in the classification process to identify the meaning of sentences made up of words. This overcomes the limitations of the Bag of Words technique, which is useful for text classification or assisting a machine in reading words in numbers. The classification performance using the Matn part is presented in Table 6.

After evaluating the Matn part, we used the Sanad part to extract the model features and ignored the Matn. The Sanad part of the Hadiths contains information about the chain of narrators, which can be useful in classifying the Hadiths. As expected, and according to Table 6, the Sanad part is also more reliable in classifying Hadiths.

To train the machine learning models, we selected appropriate hyperparameters and tuned them to obtain optimal performance. These hyperparameters are presented in Table 7.

Overall, our machine learning approach involved cleaning and preprocessing the Matn and Sanad parts, representing the Matn using TF-IDF with unigrams, bigrams, and trigrams, and training and tuning the models using the selected hyperparameters. Representing the Sanad with the chronological order of each narrator was identified by the unique number per each narrator, regardless of the different forms of the appearance of this narrator in different Hadiths. The results demonstrated that the Sanad part is more reliable in classifying Hadiths, which can be attributed to the importance of the chain of narrators in Islamic Hadith studies.

**Table 6.** Classification performance for each classifier using Matn and Sanad via classical ML methods.

| | Matn | | | Sanad | | |
|---|---|---|---|---|---|---|
| **Model** | **Precision** | **F1-Score** | **Recall** | **Precision** | **F1-Score** | **Recall** |
| Gaussian NB | 0.49268033 | 0.51502258 | 0.53948745 | 0.82207228 | 0.82173615 | 0.82140029 |
| Logistic Regression | **0.75889374** | **0.77324325** | **0.78814588** | 0.8855487 | **0.90730805** | **0.93016365** |
| SVM | 0.74899368 | 0.75825063 | 0.76773927 | **0.8873715** | 0.90588875 | 0.92519528 |
| MLP | 0.71534504 | 0.72967868 | 0.74459848 | 0.85920271 | 0.88044007 | 0.90275392 |

**Table 7.** Machine learning model hyperparameters.

| **Model** | **Hyper-Parameters** |
|---|---|
| Gaussian NB | var_smoothing = $1 \times 10^{-9}$ |
| Logistic Regression | penalty = 'l2', multi_class = 'auto' |
| SVM | kernel = 'rbf', C = 4, gamma = 0.125 |
| MLP | hidden_layer_sizes = (100), batch_size = 'auto', activation = 'relu', solver = 'adam' |

## 5. Discussion

From the experimental results, we noticed that the author classification mainly depends on Sanad. However, we have applied our models to the Sanad in addition to the Matn. We used the deep learning approach alongside the traditional machine learning approach to see which approach is better. As mentioned before, the classification problem according to the author was mainly affected by the narrators in the Sanad narration chain, and the results affirm our hypothesis.

Experiments revealed that the DL approach outperformed ML when using the Sanad part. On the other hand, the traditional ML approach is superior when it comes to the Matn part, as shown in Table 8, which is driven from Tables 5 and 6. Table 8 summarizes the best results of the DL and ML approaches.

Training a model using the narration chain is a complex process, as we need to consider multiple factors, such as the identity of the narrators, their properties and relationships, and the chronological order of each narrator. These complex features need a sophisticated model that can capture the intricate relationships between the narrators and make accurate predictions. The DL approach is well-suited for this task, as it can learn from the data and capture complex relationships using its hidden layers. Thus, if we consider the F1-score a comprehensive metric, the Transformer-based model outperforms other models using the Sanad part by a 91.54% F1 score.

In contrast, the traditional ML approach outperformed DL when using the Matn part of the Hadiths. The reason is that the Matn of Hadiths is not too long nor complex; moreover, it does not contain challenging relationships between entities that need to be modeled by the complex DL models. Hence, ML is well-suited for classification using the Matn. Consequently, we found that the logistic regression model outperformed the other models using the Matn by a 77.32% F1 score.

Furthermore, when we used the Sanad dataset, whether with a DL or ML approach, we obtained the highest accuracies compared to the Matn dataset. This confirms what we mentioned earlier, that the classification problem is mainly attributed to the Sanad and the narration chain. Therefore, our study highlights the importance of considering the Sanad part in Hadith classification and the need for sophisticated models like deep learning approaches to capture its complex relationships.

One last finding is that we tried to combine both Matn and Sanad as one entity or sample as an input for classification models, but the results were very poor. The reason is that they have completely different meanings, and it is not fruitful to be combined.

**Table 8.** Best performing classification models.

| | Matn Part | | | Sanad Part | | |
|---|---|---|---|---|---|---|
| **Approach** | **Precision** | **Recall** | **F1-Score** | **Precision** | **Recall** | **F1-Score** |
| Deep learning | Transformers | Transformers | Transformers | RNN-LSTM | RNN-LSTM | Transformers |
| | 0.64777 | 0.67832 | 0.6627 | 0.90747 | 0.93636 | **0.91536** |
| Machine learning | Logistic Regression | Logistic Regression | Logistic Regression | SVM | Logistic Regression | Logistic Regression |
| | 0.75889374 | 0.78814588 | **0.77324325** | 0.8873715 | 0.93016365 | 0.90730805 |

## 6. Conclusions and Future Work

The classification of Hadiths according to their author or origin of narration is an important research problem in the field of Hadith studies, and many studies have been conducted to address this problem. In this paper, we have introduced a novel solution to this problem by classifying Hadiths as Marfu, Mawquf, or Maqtu, and we have collected a benchmark dataset of around 29 K Hadiths, called ABCD, which we made available for other researchers to use in their studies.

In addition, we have collected and organized a dataset of around 18 K Hadith narrators, including their attributes, relationships, properties, and occurrences in different Hadiths. We have investigated both traditional ML and DL approaches to classify the Hadiths, using both the Matn and Sanad parts of the Hadiths as inputs for our models.

Our study found that the deep learning approach outperformed traditional machine learning when using the Sanad part as a feed classification, while the traditional machine learning approach performed better when using the Matn part as an input.

The Sanad part is a critical component that contains information about the chain of narrators, which can be used to determine the authenticity and reliability of the Hadith. Therefore, our study highlights the importance of considering the Sanad part in Hadith classification and the need for sophisticated models like DL approaches to capture its complex relationships. Moreover, we found that the Matn cannot classify Hadiths accurately. Also, we found that the Matn, as a simple and short text, does not require powerful models like DL-based ones, and ML worked better when the Matn.

In terms of limitations, our study is based on a dataset that is created from some, not all, Hadith books. Consequently, the dataset contains a subset of Hadiths and narrators; so, in future work, we plan to enhance our approach by adding more Hadiths and narrators from new books. This will improve the accuracy and generalizability of the developed models, and contribute to the development of more effective classification models for Hadiths.

Overall, we claim that our work provides valuable insights into the classification of Hadiths and contributes to the growing body of research in this field.

**Author Contributions:** Conceptualization, (A.R., M.T., M.A., O.S., M.E., A.A. and E.N.); methodology, (A.R., M.T., M.A., O.S., M.E., A.A. and E.N.); software, (A.R., A.A. and E.N.); investigation, (A.R., M.T., M.A., O.S., M.E., A.A. and E.N.); writing—original draft preparation, (A.R., M.T., M.A., O.S., M.E., A.A. and E.N.); writing—review and editing, (A.R., A.A. and E.N.). All authors have read and agreed to the published version of the manuscript.

**Funding:** The authors extend their appreciation to the Deputyship for Research and Innovation, Ministry of Education in Saudi Arabia for funding this research work through project number 20/18.

**Institutional Review Board Statement:** Not applicable.

**Informed Consent Statement:** Not applicable.

**Data Availability Statement:** The datasets are available from the corresponding author upon request.

**Acknowledgments:** The study was supported by a grant from the Deputyship for Research and Innovation, Ministry of Education in Saudi Arabia through project number 20/18.

**Conflicts of Interest:** The authors declare no conflict of interest.

## Glossary

| | |
|---|---|
| Daif Hadith | is a hadith which does not fulfill the conditions of the Sahih or Hassan Hadith. There is some problem in either the chain of transmission or in Hadith contents. |
| Hadith | is the narrative derived from the sayings, actions, approvals, and discussions of the Prophet Muhammad. |
| Hadith Maqtu | is a narration attributed to a Tabi'i (a successor of one of Prophet Muhammad's companions), whether it is a statement of that successor, an action or otherwise. |
| Hadith Marfu | means a Hadith attributed to the Prophet Muhammad and a companion (Sahabi) narrated it. |
| Hadith Mawquf | refers to a narration attributed to a companion (Sahabi), whether a statement of that companion, an action or otherwise. |
| Hassan Hadith | is a Hadith that has been transmitted in a continuous chain by upright narrators, but of which the exactness is less than that of a Sahih Hadith, and is devoid of irregularities or major defects. |

| | |
|---|---|
| Kutub as-Sittah (The six books) | are six classical books containing collections of Hadith written by six Sunni Muslim scholars around two centuries after Muhammad's death in the ninth century CE. |
| Madhhab | is a school of thought within Fiqh (Islamic jurisprudence). The major Sunni madhhabs are Hanafi, Maliki, Shafi'i, and Hanbali. |
| Matn | is the actual wording of Hadith's text. |
| Mawdoa Hadith | or fabricated Hadiths are those that have been made up by someone. Not only is there no clear chain of transmission, but some people have admitted to fabricating Hadiths. |
| Mosnad Imam Ahmad | is a collection of Hadiths compiled by Imam Ahmad ibn Hanbal. It is one of the most famous and important collections of reports of the Sunnah of the Prophet Muhammad. It contains approximately 28,199 Hadith sections based on individual companions. |
| Musannaf of Abd al-Razzaq | is an early Hadith collection compiled by the eighth-century Yemeni scholar ʿAbd al-Razzaq al-Sanʿani. |
| Sahih Al-Bukhari | is a collection of Hadith compiled by Imam Muhammad al-Bukhari. The overwhelming majority of Muslims consider his collection to be the most authentic collection of reports of the Prophet Muhammad's Sunnah. It contains over 7500 Hadith in 97 books. |
| Sahih Hadith | is one with a connected chain of transmission, with each narrator being upright in character, meticulous in his narration, and reliable in his transmission. |
| Sanad | is the chronological list of people who transmitted the Matn of Hadith. |
| Sunan at-Tirmizi | is a collection of Hadith compiled by Imam Abu 'Isa Muhammad at-Tirmidhi. His collection is widely regarded to be one of the six canonical collections of Hadiths (Kutub as-Sittah) of the Sunnah of the Prophet. It contains roughly 4400 Hadiths in 46 books. |
| Sunan Ibn Majah | is a collection of Hadiths compiled by Imam Muhammad bin Yazid Ibn Majah al-Qazvini. It is usually regarded as the sixth of the six canonical collections of Hadiths (Kutub as-Sittah) of the Sunnah of the Prophet. It consists of 4341 Hadiths in 37 books. |

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
