# Peer review of "Hadiths Classification Using a Novel Author-Based Hadith Classification Dataset (ABCD)"

_2504-2289, doi:10.3390/bdcc7030141_

Round 1
Reviewer 1 Report (Previous Reviewer 2)
All review comments are revised,
None
Author Response
We responded to all comments in Round #1 of the revision and the reviewer has no extra comments.
Reviewer 2 Report (New Reviewer)
There are several areas that require further clarification and improvement.
-
Dataset Description and Justification: The paper could benefit from a more detailed description of the ABCD dataset. While the paper provides an overview of the dataset, it would be helpful to include more information about how the Hadiths were selected and why these specific Hadiths were chosen. Additionally, the authors should provide a justification for the size of the dataset and discuss whether it is representative of the broader body of Hadith literature.
-
Clarification of Performance Metrics: The authors should provide a more detailed explanation of the performance metrics used in the study. While precision, recall, and F1 score are standard metrics, the authors should explain why these specific metrics were chosen and how they are relevant to the specific task of Hadith classification. Additionally, they should discuss any limitations or potential biases associated with these metrics.
-
Explanation of Model Selection: The authors should provide more information on why they chose to use the specific deep learning and machine learning models in their study. They should discuss the strengths and weaknesses of these models and why they are suitable for the task of Hadith classification. This will provide readers with a better understanding of the methodology and the rationale behind the model selection.
-
Ethical Considerations: Given the religious nature of the texts being analyzed, the authors should discuss any ethical considerations associated with their research. This could include potential misinterpretations or misclassifications of the Hadiths, and how these could impact the religious community. They should also discuss how they ensured the respectful and sensitive handling of these religious texts in their research.
-
Discussion of Results: The authors should provide a more detailed discussion of their results. This should include an analysis of why the deep learning approach outperformed traditional machine learning when using the Sanad part as a feed classification, and why the traditional machine learning approach performed better when using the Matn part as an input. The authors should also discuss any limitations of their study and potential areas for future research.
-
Citation of Relevant Literature: The authors should consider citing the following paper: doi.org/10.1177/01655515221087663. This paper discusses similar themes and could provide valuable context and support for the authors' research.
In conclusion, while the paper is a valuable contribution to the field, addressing these points will strengthen the paper and provide a more comprehensive understanding of the research.
The quality of the English language in the paper is generally good. The authors have clearly articulated their ideas and presented their research in a coherent and understandable manner. However, there are a few instances where the sentence structure could be improved for better readability. It is recommended that the authors proofread the paper or have it reviewed by a native English speaker to ensure clarity and grammatical correctness.
Round 2
Reviewer 2 Report (New Reviewer)
The authors have successfully answered to all my questions.
This manuscript is a resubmission of an earlier submission. The following is a list of the peer review reports and author responses from that submission.
Round 1
Reviewer 1 Report
This paper intents to address the problem of identifying the author of a specific Islamic hadith, proposes a new author-based hadith classification dataset (ABCD) and introduces a model that uses machine learning and deep learning techniques to solve the classification problem. It was verified that the hadith author classification problem mainly depends on Sanad. The Sanad and Matn datasets were trained using machine learning and deep learning methods.
1. The proportion of content distribution in this paper is inappropriate. Most papers describe how to classify and make data sets in hadith. There is no specific experimental plan or process; only some are mentioned in other parts.
2. Need for more innovation. The contribution of this paper is to create a new data set. In terms of experimentation, the original model is only applied without improvement and improvement on the original basis. In this regard, there need to be more innovations.
3. What the paper wants to solve is to determine the author of a hadith. However, the paper mainly talks about the classification of hadith, and the expression logic of the experimental results is unclear. The expression in the whole paper could be more concise. There are Lots of repetitive sentences. The thinking needs to be clarified.
4. Editing errors. The dataset is named "ABCD," and the last sentence of the Abstract is named "ABDC."
5. Format problem. Figure 6 The cleaned sentences of hadith are not shown.
Overall, the paper still needs to meet the requirements for publication.
Reviewer 2 Report
1. The authors might a professional English editor to revise their manuscripts.
2. The abbreviations should have their full texts when they appear first in the manuscript.
3. The “section 3” in line 100 should be “Section 3”; the “section 4” and “section 6” in line should be “Section 4” and “Section 6”.
4. The authors should use the same citation format to cite literature.
5. Regarding Figure 2, the authors should explain how they select the suitable Hadith book for their study in detail and what is the content format of the “Hadith Classification Dataset”.
6. The title of Figure 2 should be right after Figure 2.
7. The title of Figure 3 should not contain an extra explanation; the explanation should be arranged in the legend of Figure 3.
8. The title of Figure 4 should be right after Figure 4.
9. The composition from line 221 to line 235 looks weird; please fix it.
10. The title of Table 1 looks weird; the table title does not match the table content.
11. Figure 6 does not show the cleaning Matn of a Hadith.
12. The authors should explain how they normalize Sanad and clean Matn in detail.
13. The title of Table 5 should not contain an extra explanation; the explanation should be arranged in the legend of Table 5.
14. The authors should have more explanations about the content of Table 5.
15. The authors only describe the two classification schemes, the deep learning-based classification and the machine learning based classification, they used to classify Sanad and Matn; however, they did not explain how they implement these two classification schemes. The authors should describe how they implement these two classification schemes in detail.
16. The titles of the columns in Tables 5, 6, and 7 with “F1” should have the same title name.
17. There are two Table 7 in the manuscript. The authors should fix this mistake.
18. The authors should explain how they got the content of Table 7 (the second Table 7).
19. There is no [13] in the manuscript.
20. The title of the appendix should be “Appendix – Glossary”, instead of “Glossary Appendix”.
